# Proteomic and Transcriptomic Patterns during Lipid Remodeling in *Nannochloropsis gaditana*

**DOI:** 10.3390/ijms21186946

**Published:** 2020-09-22

**Authors:** Chris J. Hulatt, Irina Smolina, Adam Dowle, Martina Kopp, Ghana K. Vasanth, Galice G. Hoarau, René H. Wijffels, Viswanath Kiron

**Affiliations:** 1Faculty of Biosciences and Aquaculture, Nord University, PB 1490, 8049 Bodø, Norway; irina.smolina@nord.no (I.S.); martina.kopp@nord.no (M.K.); ghana.k.vasanth@nord.no (G.K.V.); galice.g.hoarau@nord.no (G.G.H.); kiron.viswanath@nord.no (V.K.); 2Department of Biology, Bioscience Technology Facility, University of York, York YO10 5DD, UK; adam.dowle@york.ac.uk; 3Bioprocess Engineering, AlgaePARC, Wageningen University, 6700 AA Wageningen, The Netherlands; rene.wijffels@wur.nl

**Keywords:** proteomics, transcriptomics, *Nannochloropsis*, EPA, TAG, phosphorus, nitrogen, bioreactor

## Abstract

Nutrient limited conditions are common in natural phytoplankton communities and are often used to increase the yield of lipids from industrial microalgae cultivations. Here we studied the effects of bioavailable nitrogen (N) and phosphorus (P) deprivation on the proteome and transcriptome of the oleaginous marine microalga *Nannochloropsis gaditana*. Turbidostat cultures were used to selectively apply either N or P deprivation, controlling for variables including the light intensity. Global (cell-wide) changes in the proteome were measured using Tandem Mass Tag (TMT) and LC-MS/MS, whilst gene transcript expression of the same samples was quantified by Illumina RNA-sequencing. We detected 3423 proteins, where 1543 and 113 proteins showed significant changes in abundance in N and P treatments, respectively. The analysis includes the global correlation between proteomic and transcriptomic data, the regulation of subcellular proteomes in different compartments, gene/protein functional groups, and metabolic pathways. The results show that triacylglycerol (TAG) accumulation under nitrogen deprivation was associated with substantial downregulation of protein synthesis and photosynthetic activity. Oil accumulation was also accompanied by a diverse set of responses including the upregulation of diacylglycerol acyltransferase (DGAT), lipase, and lipid body associated proteins. Deprivation of phosphorus had comparatively fewer, weaker effects, some of which were linked to the remodeling of respiratory metabolism.

## 1. Introduction

Bioavailable nitrogen and phosphorus are essential macronutrients required by microalgae for optimal, balanced growth. In the oceans, the effects of nitrogen (N) and phosphorus (P) supply on phytoplankton physiology and elemental stoichiometry are well recognized [1,2], where nutrient abundance often controls primary production, community structure, and ultimately the flux of matter and energy through ecosystems [3,4]. Many species of microalgae also have applications in biotechnology, where modulating the nutrient supply to intensive cell cultures is a common technique used to induce the accumulation of triacylglycerol (TAG) and secondary carotenoids [5,6]. Understanding how microalgae respond to changes in nutrient availability, especially the supply of N and P, is therefore valuable for characterizing their behavior in natural and industrial settings.

Protein accounts for a large share of cellular N, but nitrogen is also a component of nucleic acids (RNA and DNA) and chlorophyll. Phosphorus is required in lower amounts, but is nevertheless embodied in nucleic acids, phospholipids, post-translational modifications (e.g., phosphoproteins), and ATP [7,8,9]. Though N and P are often found in the same molecules, the effects of their abundance on microalgae physiology can be profoundly different. Nitrogen deprivation typically leads to substantial reductions in growth, protein and chlorophyll content, concomitant with increased neutral lipids, carbohydrates, or secondary carotenoids, depending on the species. The effects of P- deprivation are often more subtle, but have been consistently linked to remodeling of the lipid profile [10,11], where phosphorus-containing lipid classes are substituted for nonphosphorus lipids [9]. The active remodeling of the microalgae cell under N and P stress implicates the roles of a large number of regulatory pathways, but we still lack a deep understanding of the molecular mechanisms at work.

Transcriptome-based studies have identified patterns of gene expression during nutrient stress response and product formation [12,13]. However, eukaryotic microalgae have evolved through diverse endosymbiotic routes, and different families, genera, and species may respond differently to similar treatments. Quantitative transcript sequencing can imply that gene expression directly regulates the abundance of proteins, yet there is often only moderate association between mRNA and protein expression [14,15]. For example, studies on human cell lines have found low correlation (R^2^ = 0.22–0.29) between mRNA and protein measurements [16,17], although stronger relationships have been reported from mouse cells (R^2^ = 0.41), bacteria (R^2^ = 0.47), and yeast (R^2^ = 0.58) [16,18]. One explanation for this is the variable role of post-transcriptional mechanisms in different organisms and conditions [15,18,19]. Compared to transcriptomics, then, proteomics should provide more direct measurement of metabolic activity inside the cell, but such studies in microalgae are relatively few. Key questions include, how does macronutrient supply reshape the algal proteome, and do proteomic and transcriptomic methods describe similar metabolic patterns?

The marine eustigmatophyte *Nannochloropsis* is one of a handful of industrially tractable oleaginous microalgae. Its ~30 Mbp haploid nuclear genome is compact, containing around ten-and-a-half thousand protein coding genes, varying slightly amongst the assemblies of different strains [20,21]. Despite its modest size, the *Nannochloropsis* genome encodes a disproportionately large number of genes involved in lipid synthesis, including 11 or more copies of diacylyglycerol acyltransferase-2 (DGAT2), which performs the terminal step in TAG synthesis via the Kennedy pathway [20,22]. Under adverse conditions, especially N starvation, *Nannochloropsis* can accumulate substantial quantities of TAG in oil bodies, reaching 50% or more of the cell dry mass [23]. *Nannochloropsis* is also remarkable as a genus that can synthesize large amounts of the long-chain polyunsaturated fatty-acid C20:5*n*-3 (eicosapentanoic acid or EPA), which is highly valued in human and animal diets [24,25].

Here we used flat-plate photobioreactors operated as turbidostats to selectively apply nitrogen and phosphorus deprivation to *Nannochloropsis gaditana*. The molecular patterns emerging under N and P deficient conditions were characterized using Tandem Mass Tag (TMT) based quantitative proteomics and are supported by transcriptome (mRNA) sequencing of the same samples. Our analysis first examines the global (cell-wide) patterns of protein and transcript abundance, before exploring the primary effects of N and P starvation on the subcellular proteomes, gene clusters, and metabolic pathways. Individual pathways and proteins that were either highly impacted, or relevant to biotechnology applications, are investigated and discussed.

## 2. Results

### 2.1. Turbidostat Cultivation Dynamics, Lipids, and Fatty-Acids

Control cultures were maintained in nutrient-replete, steady-state conditions throughout the experiments with a specific growth rate of 0.55 ± 0.07 d^−1^ and a cell density of 2.6 ± 0.3 g L^−1^. In nitrogen (N-) and phosphorus (P-) deprived treatments the growth rates declined, but other variables inside the bioreactor including the average light intensity, were largely maintained (Figure 1a). In the N- and P- cultures, either nitrate or phosphate was exhausted within 28 h due to rapid nutrient uptake coupled with high biomass turnover and dilution with fresh medium (Figure 1a). Nitrate-starved cultures showed a gradual increase in cell density toward the end of the experiment, a result of maintaining constant turbidity whilst the cells experienced chlorosis (loss of pigmentation). The N- cultures experienced an immediate reduction in growth rate to 0.11 ± 0.02 d^−1^ at day 3 and 0.05 ± 0.02 d^−1^ at day 5. In comparison the onset of P- conditions was more dampened with the growth rate 0.49 ± 0.06 d^−1^ at day 3 and 0.44 ± 0.07 d^−1^ at day 5. Analysis of fatty-acids showed a substantial increase in TAG comprised primarily of C16:0 and C16:1 fatty-acids in the N- treatments (Figure 1b). After 5 days in N- conditions, fatty-acids in TAG comprised 21.4% of the cell dry weight but remained at only 1.0% and 2.2% of the dry weight in the control (C) and P- treatments, respectively. The long-chain PUFAs eicosapentanoic acid (EPA, C20:5*n*-3) and arachidonic acid (ARA, C20:4*n*-6) were mostly present in the polar lipids. At day 5 the EPA accounted for 26.5% and ARA for 2.5% of total fatty acids (TFA) in control cultures. In N- cultures the EPA content was reduced substantially to 6.3% TFA after 5 days, due to the reduction of polar lipids and the accumulation of fatty acids in TAG.

### 2.2. Identification and Differential Expression of Proteins and Their Transcripts

In total 3423 proteins were identified across all of the tested conditions. After 3 days of N- deprivation 1543 of these proteins were significantly differentially regulated, whilst in P- treatments only 113 proteins were significantly differentially regulated (Figure 2). Transcriptome analysis showed that after 3 days of N- treatment, 1448 of the 10,496 genes in the B31 genome were differentially expressed, where 528 transcripts were upregulated and 920 were downregulated. After 5 days of N- treatment, the number of differentially expressed genes (DEGs) increased to 2371, where 859 were upregulated and 1512 were downregulated. Phosphorus depletion resulted in far fewer DEGs, where only 52 genes were upregulated and two were downregulated after 3 days, increasing to a total of 122 DEGs after 5 days. Principal components analysis showed that in the protein dataset there was distinct clustering of N- samples, but much weaker demarcation between P- and control treatments (Appendix A). Principal components analysis of the transcriptomic data indicated clear divergence between each of the treatments after 3 days, strengthening further after 5 days.

### 2.3. Correlation between the Nannochloropsis Proteome and Transcriptome

The global patterns in protein and mRNA abundance were examined using three complimentary approaches. First, the correlation between the log_2_ fold changes (L_2_fc) of mRNA transcripts and their corresponding proteins was performed (Figure 3a). The N-/C treatment yielded moderate correlation (R^2^ = 0.25), whilst the correlation in P-/C treatments was much weaker (R^2^ = 0.08). Our second method combined data for all observations (C, N-, and P- treatments) together, and a linear mixed-effects model was used to describe the relationship between mRNA abundance (log (RPKM)) and protein abundance (log (Mol%)) across all gene/protein accessions (Figure 3b). For comparative purposes, a conventional Pearson’s R^2^ of 0.31 was also calculated for the same data, indicating moderate positive correlation between transcript and protein abundance. Our third method fitted individual linear regression models to each gene/protein pair, yielding 2576 regression models. The distribution of R^2^ values from these linear models are presented in Figure 3c (upper panel), and for only the subset of proteins which showed significant differential expression (Figure 3c lower panel). The median R^2^ for all accessions was 0.29, but increased substantially to R^2^ = 0.58, with a shoulder at R^2^~0.8, when only the significantly differentially expressed proteins were included. For those significantly differentially expressed proteins, 79% of the gene/protein correlation slopes were positive, the remaining 21% were negative (Figure 3d). Together, these three alternative approaches characterize a moderate but detectible cell-wide association between mRNA and protein expression in these data.

### 2.4. The Effect of Nitrogen and Phosphorus Stress on Subcellular Proteome Remodeling

To investigate large-scale changes in subcellular proteomes under N- and P- conditions, we examined the overall fold changes of proteins after grouping them into their respective cellular locations. For most compartments, N- treatments exhibited greater variance in protein abundance than P- treatments (Figure 4). Proteins associated with the plastid were mostly downregulated under nitrogen deprivation, with a median L_2_fc of −0.42. Proteins localized to the mitochondrion, membranes and the endoplasmic reticulum (ER) also displayed variation in L_2_fc, but their median fold changes each remained around zero (L_2_fc 0.00, 0.02, and −0.08, respectively). The data indicate that under N- conditions the plastid proteome shrank, whilst the ER, mitochondrial and membrane proteins were remodeled but did not substantially change overall size. In P- treatments there were no substantial shifts in expression of any of the subcellular proteomes, and variation in L_2_fc was much lower than those in N- treatments, indicating only limited remodeling.

### 2.5. Functional Enrichment Analysis of Differentially Expressed Proteins and Transcripts

To capture the main patterns in gene expression and protein abundance, gene ontology (GO) and KEGG pathway ontology (KO) terms were examined (Figure 5 and Figure 6). Under N- conditions changes in the proteome and transcriptome were mostly concordant, where downregulation of proteins and mRNA transcripts was observed in protein translation processes (GO:0006412), protein-chromophore linkage (GO:0018298), and light-independent chlorophyll biosynthesis (GO:0036068), together with photosynthesis (GO:0015979) and its light-dependent (GO:0009765) and light-independent reactions (GO:0019685). Fewer gene and protein GO categories were significantly upregulated in N- treatments, but genes and proteins with roles in amine metabolism (GO:0009308), the tricarboxylic acid cycle (GO:0006099), and nucleotide catabolism (GO:0009166) were increased.

In P- treatments, over-represented GO terms for proteomic and transcriptomic data were less concordant. The downregulation of proteins involved in translation (GO:0006412), protein stabilization (GO:0050821), D-ribose catabolic process (GO:0019303), and carbohydrate transport (GO:0008643), together with the upregulation of tricarboxylic acid cycle (GO:0006099) and glycolytic process (GO:0006096), was not echoed by the transcriptome (Figure 5). After 5 days of phosphorus starvation, gene expression associated with tRNA (GO:0006418) and rRNA processing (GO:0006364) were also lowered, together with reductions in ribosome biogenesis (GO:0042254), ribosome assembly (GO:0000028), protein refolding (GO:0042026), and amino-acid biosynthesis (GO:0008652). Transcripts associated with amine metabolism (GO:0009308) were also downregulated after 5 days of P deprivation, contrasting with the upregulation of the same group during N deprivation. Upregulated gene clusters in P- treatments included increases in phosphate-ion transport (GO:0006817) and increases in transcripts associated with lipid catabolism (GO:0016042), ATP synthesis (GO:0015986, GO:0042773), and oxidative phosphorylation (GO:0006119).

In nitrogen-starved cells, KEGG pathways related to photosynthesis (KO:00195) and ribosomes (KO:03010) were downregulated in both proteomic and transcriptomic data. (Figure 6). Under P- conditions proteins in the KEGG pathways glycolysis/gluconeogenesis (KO:00010), the TCA cycle (KO:00020), and oxidative phosphorylation (KO:00190) were upregulated. However, these increases in respiration-associated protein groups were not mirrored by the transcriptome. Instead, after 5 days transcriptome data indicated downregulation of several pathways linked to lysine biosynthesis (KO:00300) and aminoacyl-tRNA biosynthesis (KO:00970), implying reduced translation activity under protracted P-deprivation.

### 2.6. Translation, Nitrogen Acquisition, and Metabolism

Under N- conditions, 12 of the 30 most downregulated proteins were ribosomal (Table 1), mostly 30S and 50S that are plastid-associated. The L_2_fc of all ribosomal proteins were examined, and we found that both plastidic ribosomes and ribosomal proteins of eukaryotic origin (40S and 60S) were downregulated after 3 days of N- conditions (Appendix A). In P- treatments the expression of ribosomal proteins and their transcripts was not substantially changed. Both nitrate and nitrite reductase were among the most downregulated proteins in the N- treatments, highlighting the reduced investments in N acquisition from the extracellular environment.

The reduced plastid proteome and diminished photosynthetic capacity associated with N starvation led us to hypothesize that enzymes involved with protein/amino-acid catabolism, nitrogen recycling, and recovery could be upregulated. Consistent with increases in amine metabolic processes (GO:0009308, Figure 5), an amine oxidase (W7TFN3_9STRA) was the second-most upregulated protein under N- conditions with an L_2_fc of +1.38 (Table 1). In P- treatments, the same protein was significantly downregulated (L_2_fc −0.32, *p* < 0.001). Further searching through the proteome revealed an additional six proteins annotated as amine oxidases, and of these a further two were significantly upregulated under N- conditions (Appendix A). Additional proteins associated with amine metabolism were also significantly upregulated in N- treatments, including an amine dehydrogenase (W7TI92_9STRA) with an L_2_fc of +0.77.

### 2.7. Tricarboxylic Acid (TCA) Cycle, Glycolytic Processes, and Oxidative Phosphorylation

Evidence from Figure 4, Figure 5 and Figure 6 indicated that remodeling of mitochondrial or respiratory activity took place under both N- and P- conditions. To establish which proteins and transcripts were differentially expressed, and how regulatory activity potentially differed under N- and P- conditions, the L_2_fc of respiratory-associated proteins were examined together with their transcripts (Figure 7). In N- conditions, most proteins and transcripts associated with the TCA cycle were upregulated, but those associated with glycolytic processes were both up- and downregulated. Two glycolytic enzymes, glyceraldehyde-3-phosphate dehydrogenase and phosphoglycerate kinase included multiple copies that were not coregulated with one another, with different accessions showing divergent patterns of regulation (e.g., W7U208_9STRA vs. W7T2R0_9STRA). In P- conditions, most TCA cycle and glycolytic proteins and transcripts were weakly upregulated.

### 2.8. Fatty-Acid and Acyl-CoA Metabolism

An Acetyl-CoA carboxylase protein (I2CQP5_NANGC) was significantly upregulated during P-starvation (L_2_fc +0.12, *p* < 0.001), but significantly downregulated under N- conditions (L_2_fc −0.50, *p* < 0.001). Two proteins annotated as Acyl CoA synthetase were identified, but only one long-chain Acyl-CoA synthetase (LACS, W7TGG5_9STRA) was significantly upregulated under N- conditions (L_2_fc +0.36, *p* < 0.001).

### 2.9. Polyunsaturated Fatty Acid (PUFA) Metabolism

The primary route to medium and long-chain polyunsaturated fatty-acid biosynthesis in microalgae is via a series of steps involving desaturase and elongase enzymes. A ∆5 desaturase (K8YSX2_NANGC) was amongst the most downregulated proteins in N- treatments (Table 1). Six other desaturase enzymes were also significantly downregulated during N- conditions (Appendix A), including a Δ12 ω-6 desaturase (K8YR13_NANGC) and a glycerolipid ω-3 desaturase (I2CR09_NANGC), with L_2_fc of −0.37 and −0.53 respectively (*p* ≤ 0.005). Under P- conditions the abundance of the same Δ5, Δ12, and glycerolipid desaturases did not significantly change.

### 2.10. Proteins Associated with TAG Biosynthesis and Storage in Oil Bodies

The most upregulated protein in N- treatments with an L_2_fc of +1.93 (*p* < 0.001) was a lipid droplet surface protein (W7TWF7_9STRA), which is concordant with the substantial increases in TAG observed in the same samples (Table 1, Figure 1). Although the *N. gaditana* genome is reported to encode 11 copies of DGAT2, only one diacylglycerol acyltransferase (DGAT) family protein (W7U9S5_9STRA) was identified. This protein was significantly upregulated under N- conditions (L_2_fc +0.30, *p* = 0.004), but not under P- conditions (L_2_fc −0.14, *p* = 0.420). In comparison, the transcript data quantified the expression of eight different genes annotated as DGAT or DGAT2, where three were significantly upregulated under N- conditions and two were significantly downregulated (Appendix A). Further upstream in lipid biosynthesis, Lysophosphatidylglycerol acyltransferase (LPAT) catalyzes the conversion of lysophosphatidic acid to phosphatidic acid. We identified a single LPAT protein (K8YP17_NANGC), that did not respond significantly in either N- or P- conditions.

### 2.11. Glycerolipid and Phospholipid Biosynthesis

A single protein annotated as monogalactosyldiacylglycerol synthase (MGDG synthase, W7TN13_9STRA) was not significantly differently expressed in either N- or P- conditions (L_2_fc < 0.07, *p* > 0.130). A choline/ethanolamine kinase family protein (K8YV04_NANGC) was significantly upregulated (L_2_fc +0.28, *p* = 0.001) in P- conditions, but was not significantly changed in N- conditions (L_2_fc +0.13, *p* = 0.072). The proteomics data also identified a Udp-sulfoquinovose synthase (W7TMH8_9STRA) that was significantly downregulated in N- conditions (L_2_fc −0.2, *p* < 0.001), but significantly upregulated in P- conditions (L_2_fc +0.24, *p* < 0.001). In P- conditions an Acid sphingomyelinase-like phosphodiesterase 3b (W7TQ09_9STRA) was amongst the most upregulated proteins with an L_2_fc of 0.68 (*p* = 0.011) (Table 2).

### 2.12. Lipase Activity and Lipid Catabolism

In P- conditions a single lipase (W7TUB0_9STRA) was significantly downregulated (L_2_fc −0.32, *p* = 0.001). The same accession was substantially upregulated under N- conditions (L_2_fc +1.06, *p* < 0.001), in addition to the significant upregulation of five other lipase family proteins, including two lysophospholipases (Appendix A).

### 2.13. Polyketide Synthase, Fatty Acid Synthase, and Lipoxygenase Expression

Six proteins annotated as polyketide synthases (PKS) were detected in the proteomics data, but none responded significantly in either the nitrogen-starved or phosphorus-starved treatments (Appendix A). A single fatty acid synthase (FAS1) domain protein (W7TBQ5_9STRA) was significantly downregulated in nitrogen-starved conditions (L_2_fc −0.47, *p* < 0.001) but not phosphorus-starved conditions (L_2_fc −0.10, *p* = 0.091). An Arachidonate 5-lipoxygenase (K8Z8I5_NANGC) was also amongst the most upregulated proteins with an L_2_fc of +0.71 (Table 1), whilst a manganese lipoxygenase protein (W7TYD4_9STRA) was also significantly upregulated under N- conditions, providing evidence for the upregulation of oxylipin pathways during nitrogen starvation.

## 3. Discussion

The 3423 proteins identified in this study represent a third of the gene models in the *N. gaditana* genome [20,21] providing deep profiling of the *Nannochloropsis* proteome. The data also offers the opportunity to compare the expression of proteins with their mRNA transcripts.

### 3.1. Global Correlation of Nannochloropsis Protein and Transcript Expression

Integrating different ‘omics datasets is a challenge but offers the chance to ask valuable questions. On one hand, transcriptome sequencing provides high-throughput measurements of global responses to physiological stress and has been widely adopted. Nevertheless, the abundance and activity of proteins in cells, which ultimately determines the phenotype, is regulated by numerous mechanisms beyond mRNA expression alone [18]. Our proteomic and transcriptomic data presented here are concordant with studies on other organisms, where generally only weak-moderate associations have been observed at the whole-cell level. Whether the unexplained residual variation is due to post-transcriptional mechanisms or to methodological sensitivity, is not always clear [15].

Correlating the L_2_fc (Figure 3a) is a straightforward method of associating transcript and protein data that relies only on relative changes in expression. Here N starvation produced a stronger correlation than P starvation, likely due to larger changes in protein and transcript abundance under N stress. However, our additional correlation methods help to provide a more complete picture. In Figure 3b we used measures of protein and transcript abundance, rather than their relative fold changes, and obtained an R^2^ = 0.31. This value is comparable to observations in the model plant *Arabidopsis thaliana* (R^2^ = 0.27–0.46) and bacteria (R^2^ = 0.20–0.47), but lower than yeasts (R^2^ = 0.34–0.87) [16]. When individual linear models were fitted separately to data from each protein/transcript, we were able to show the heterogeneity of correlations across different genes (Figure 3c). Proteins that were significantly differentially expressed often exhibited higher correlation with their transcripts, providing support for the role of effect-size in determining the strength of gene–protein correlations. Nevertheless, a proportion of significantly regulated proteins remained only weakly correlated with their transcripts. Like other eukaryotes, microalgae employ a multitude of post-transcriptional systems, but to what extent ncRNAs, splicing, post-translational modifications, and protein turnover [19,26,27,28,29] impact transcript/protein/metabolome relations in oleaginous microalgae, is not yet very clear. The effect of N, but not P deprivation, on reducing ribosomal protein abundance illustrates that ribosome density varies with certain stress responses, representing a further layer of regulation between transcription and translation. Lastly, the dynamic nature of gene–protein regulatory circuits may be a critical variable [30]. Our turbidostat cultures controlled for the light intensity, but during the experimental treatments the cultures remained non-steady-state systems, where there may be overshoot in the transcriptional control of protein abundance [30,31]. Future studies can address this aspect by using alternative bioreactor control strategies.

### 3.2. N and P Deprivation Remodels Organelle Proteomes and Energy Metabolism

Eukaryotic cells are highly compartmentalized and the size, spatial arrangement, and contacting of subcellular compartments is re-optimized under stress conditions. In our data the dampened onset of P- stress contrasted with the rapid reduction in growth and changes in protein/gene expression observed in N- conditions. These differences can be reconciled by the way phosphorus is utilized inside the cell. Under nutrient-replete conditions luxury phosphorus uptake takes place and cells can accumulate excess reserves of intracellular phosphorus which acts as a short-term buffer during P- conditions [32]. Secondarily, certain classes of phosphorus-containing compounds can be functionally replaced by phosphorus-free alternatives (see Section 3.3), which reduces the impact of P- conditions on metabolism.

Our analysis showed that the plastid proteome was downregulated under N- conditions, consistent with the nitrogen-starved phenotypes (chlorosis and reduced polar lipid content) and the downregulation of mRNAs and proteins associated with photosynthesis. Despite the reduced photosynthetic capacity, mitochondrial proteins remained on average at comparable abundance to the control treatments, but there was evidence of reorganization, The changes in TCA cycle and glycolytic proteins under N- and, weakly, under P- conditions, highlights the active role played by respiratory processes during macronutrient stress. Previous research has indicated increased expression of glycolytic enzymes including glyceraldehyde-3-P dehydrogenase during N- conditions [33]. Our data indicates that these proteins, which are present in multiple copies, can show opposing patterns of regulation and therefore more information e.g., on cellular localization and targeting is required before their roles can be fully understood. In plants, phosphate deprivation is associated with regulation of alternative pathways in glycolysis and oxidative phosphorylation [34], and evidence from the proteome of the diatom *Phaeodactylum* [35] also indicates upregulation of TCA cycle activity under N-limited conditions. As mitochondrial activity is central to pathways in energy metabolism and amino acid cycling, alternative configurations of the mitoproteome play a central role in acclimation to protracted macronutrient deficits and further research is needed on mitochondrial metabolic flux under nutrient stress.

### 3.3. Lipid Metabolism and Remodeling

The regulation of lipid metabolism in oleaginous microalgae has been the subject of substantial scientific and commercial attention, yet the underlying mechanisms are still not completely resolved [36]. Transcriptome sequencing studies have shown that the genes involved in lipid biosynthesis are actively regulated during nutrient-induced stress [12], yet attempts to increase oil yields by overexpression of key genes have yielded mixed results [37], indicating that lipid biosynthetic enzymes are not necessarily rate-limiting. In *Nannochloropsis*, nitrogen starvation is primarily associated with TAG production and lipid storage in oil droplets, but surprisingly our GO and KEGG enrichment analysis (Figure 5 and Figure 6) did not prioritize lipid-related protein or gene families during oil accumulation. However, several lipid-related proteins were strongly upregulated under N starvation, including a lipid droplet surface protein (LDSP) with the highest fold change in the whole dataset. Similar proteins have been characterized from *Nannochloropsis oceanica*, *Chlamydomonas*, and *Phaeodactylum* [38,39]. These proteins play a structural role in oil bodies, and so their abundance scales with neutral lipid accumulation [39].

The *N. gaditana* genome is reported to encode 11 DGAT2 genes, but we were only able to distinguish one diacylglycerol acyltransferase protein, although the expression of eight different DGAT2 genes were counted in the transcript data. The upregulation of DGAT under N- stress, but not under P- stress, indicates a regulatory role in TAG accumulation, and the same accession (corresponding to gene Naga_100006g86) also responds to changing light conditions in this species [40]. We identified a single LPAT protein that was unresponsive to either N- or P- conditions. However, several LPAT orthologs are present in the *Nannochloropsis* genome and their subcellular localization and functional role is not shared equally among them [36]. The protein Acetyl-CoA carboxylase, which drives lipid biosynthesis in the plastid [41], was downregulated under N-deprived conditions indicating reduced de-novo fatty acyl chain biosynthesis.

Macronutrient deprivation not only induces accumulation of TAG, but the remodeling of membrane (polar) lipids. Nitrogen deprivation especially induces the degradation of plastidic glycerolipids, especially phosphatidylglycerol (PG), monogalactosyldiacylglycerol (MGDG), and digalactosyldiacylglycerol (DGDG) that that contain the majority of the EPA [42]. The fate of PUFAs under nutrient stress has important consequences for the lipid and fatty acid composition of the cell, and different processes including de-novo PUFA synthesis, translocation, and degradation/oxidation of fatty acids together contribute to the overall lipid profile. Recent evidence indicates that limited de-novo synthesis of LC-PUFAs does occur during nutrient deprivation [43], but the degradation of polar lipids and the translocation of PUFAs into TAG are significant processes that can affect the nutritional properties of microalgae. We found that PUFA biosynthesis was strongly downregulated in N- conditions, with major reductions in desaturase activity. In *Nannochloropsis*, Δ5 desaturase activity is associated with ARA and EPA biosynthesis [23], and together the proteomic data and fatty-acid profiles indicate that de-novo LC-PUFA biosynthesis probably plays only a minor role in lipid composition under N starvation.

Lipid-class remodeling has been associated with phosphorus starvation, where specific classes of P-containing membrane lipids are substituted with nonphospholipids [10]. Phospholipid remodeling in plants and microalgae involves acyltransferase and phospholipase activity [44]. Whilst various proteins annotated as phospholipases were identified in our data, none were significantly upregulated under P- conditions. Instead, increased lipase activity was a signature of oil-accumulating cells under N- starvation. However, we found that a choline/ethanolamine kinase was upregulated under P- conditions, which could indicate attempts to maintain phospholipid (phosphatidylcholine, PC and phosphatidylethanolamine, PE) production in these conditions. We also identified a Udp-sulfoquinovose synthase protein that was significantly downregulated in N- conditions, but significantly upregulated in P- treatments. This enzyme is associated with the synthesis of sulfoquinovosyldiacylglycerol (SQDG), a thylakoid lipid that can potentially replace and compensate for loss of phospholipids, especially PG, during phosphorus-scarce conditions [11].

An interesting feature of our data was the upregulation of two putative lipoxygenase (LOX) proteins under N- stress. Lipoxygenases provide the enzymatic route to oxylipin production where PUFAs, primarily C18 and C20 series, are converted to various oxidized lipid derivatives [45]. Oxylipins have roles in cell signaling and stress response and, although LOX activity has not been widely investigated in different microalgae species, oxylipin production has been measured in *Nannochloropsis* [46], and hydroxylated EPA was abundant in the metabolome of the diatom *Phaeodactylum tricornitum* under similar experimental conditions [35].

## 4. Materials and Methods

### 4.1. Cultivation

*Nannochloropsis gaditana* (CCMP 526, National Center for Marine Algae and Microbiota, East Boothbay, ME, USA) was cultivated in 400 mL flat plate photobiorectors (Algaemist-S, Wageningen UR, The Netherlands) using f/2 medium (Guillard and Ryther, 1962). The nutrient concentrations were increased proportionally to support high cell density, equivalent to 3.0 g L^−1^ NaNO_3_. Cultures were maintained as turbidostats (constant optical density) by automatically adding fresh medium and collecting the overflowing broth. Turbidostat cultures provide a high level of experimental control by eliminating variables such as changes in internal irradiance that typically occur in batch or flask cultures. The temperature (25.0 ± 0.2 °C) was maintained by internal heating/external cooling modules and a constant irradiance of 350 μmol m^−2^ s^−1^ was provided by warm-white light emitting diodes. These conditions ensured high cell density and rapid biomass turnover. Before experimental treatments the cultures were maintained for several days, where they reached a constant growth/dilution rate. Control (C) treatments were subsequently maintained at the same steady-state, whilst nitrogen (N-) and phosphorus (P-) stress treatments were selectively applied by omitting either nitrate or phosphate from the feed medium. The high biomass turnover ensured cells in stress treatments were subjected to a rapid, natural depletion of either N or P. Since there were two photobioreactor units, the cultivation sequence was designed to avoid treatment bias (Appendix A), and in total there were *n* = 4 independent replicate cultures for C, N-, and P- conditions. Conditions inside the photobioreactors were recorded by a program written in Python v2.7, running on a Raspberry Pi single-board computer (Raspberry Pi foundation, UK). The maximum duration of our experiment was 5 days, by which time growth in N- treatments had nearly ceased and the limit of turbidity control was reached. Based on the cultivation data in Figure 1 we selected day 3 for proteomics and transcriptomics analysis, because it represented the mid-point in the onset of stress conditions, allowing sufficient time to detect metabolic and molecular changes in the cells.

### 4.2. Sample Collection

Samples for proteomic and transcriptomic analysis were each collected into 2.0 mL tubes. Cells were immediately pelleted by centrifugation (5000 rcf, 2 min) and quenched in liquid nitrogen, then stored at −80 °C. Samples for metabolite analysis were collected in 2.0 mL tubes and additionally desalted by washing with isotonic ammonium formate, then stored at −20 °C. The sample supernatant was retained for analysis of nitrate and phosphate. The sample time points selected for molecular characterization are shown in Table 3. Our experiment comprised 12 turbidostat cultivations, but only 10 TMT labels were available for proteomic analysis. Thus, control and N- proteome treatments each have four biological replicates, whilst P- treatments have two replicates for the proteome. Statistical analysis accounted for the degrees of freedom and multiple comparisons.

### 4.3. Lipid Analysis

Polar and neutral lipids were separated by solid phase extraction and the fatty acids were analyzed with a Gas Chromatograph and Flame Ionization Detector (GC-FID). Approximately 8 mg lyophilized samples were weighed with a precision balance (Mettler Toledo, Columbus, OH, USA, MX5) and transferred into 2.0 mL tubes containing 300 μL of 0.1 mm glass beads. Cell disruption was performed by adding 1.0 mL chloroform:methanol (2:2.5) spiked with C15:0 TAG (tripentadecanoin) internal standard, before bead-milling. The homogenate was transferred to a 10 mL glass tube with the addition of another 3.0 mL chloroform:methanol. Phase separation was used to recover the chloroform fraction, which was then dried under a stream of N_2_ to recover total lipids. Polar and neutral lipid extracts were then prepared using solid-phase columns (Waters Sep-Pak 6cc/1g silica) and derivatized to fatty-acid methyl-esters (FAMEs) by adding 3.0 mL of 12% H_2_SO_4_ in methanol, then heating at 70 °C for 3 h. FAMEs were separated and quantitated using a Scion 436 GC-FID (Bruker, USA) fitted with a splitless injector and a 30 m CP-WAX column (Agilent Technologies, USA). Supelco 37-component standards (Sigma-Aldrich, Oslo, Norway) were used for identification and quantitation of the FAMEs with five-point calibrations. Blanks were included throughout extraction and derivatization, to eliminate trace background peaks.

### 4.4. Nutrient Analysis

The concentration of nitrate in the broth was measured with standard colorimetric reagents using a miniaturized microplate method and NADH:nitrate reductase [25]. The absorbance was measured at 540 nm with a Tecan Sunrise microplate reader. Seven-point calibrations were included in each plate (R^2^ > 0.995). Phosphate was analyzed with the ammonium molybdate/ascorbic acid method, and the absorbance was measured at 650 nm with a 1.0 cm cell.

### 4.5. Proteomics

Protein was extracted by resuspending cell pellets in 1.0 mL of extraction buffer (phosphate buffered saline +0.03% Triton X-100 + protease inhibitor cocktail) on ice, and homogenized briefly with a bead mill (Precellys, Bertin Instruments, Montigny-le-Bretonneux, France, 0.1 mm glass beads, 6500 rpm, 15 s). The suspension was centrifuged (20,000 rcf, 15 min, 4 °C) and the supernatant transferred to new tubes. Proteins were then precipitated by adding five volumes of ice-cold acetone, followed by centrifugation (20,000 rcf, 15 min, 4 °C). The supernatant was removed, and the protein pellets were allowed to air dry for 2 min at room temperature. Protein pellets were suspended in Laemlli buffer and the protein concentration of each sample was measured in duplicate with a BCA protein assay kit (Microplate BCA™ Protein Assay Kit—Reducing Agent Compatible, Thermo Scientific, Waltham, MA, USA). A seven-point calibration was used (R^2^ > 0.999) and samples were blank-corrected using the sample buffer (Appendix A). A standardized 95.1 μg of protein from each sample was loaded to an SDS-PAGE gel and trapped for analysis.

Analysis and database searching was performed by University of York metabolomics and proteomics facility (York, UK) using 10-plex Tandem Mass Tags (Thermo Scientific, TMT10plex™). In-gel tryptic digestion was performed after reduction with dithioerythritol and *S*-carbamidomethylation with iodoacetamide. Digests were incubated overnight at 37 °C, then peptides were extracted with 50% aqueous acetonitrile containing 0.1% trifluoroacetic acid, before drying in a vacuum concentrator and reconstituting in aqueous 0.1% trifluoroacetic acid. Peptides were buffer exchanged into aqueous 50 mM triethylammonium bicarbonate using Strata C_18_-E cartridges before TMT labelling (Appendix A for label-sample assignments). Labelled samples were combined together, loaded onto a conditioned reversed-phase C_18_ spin column (Pierce) and subject to centrifugation at 5000 rcf for 2 min before washing with 300 μL of LC-MS grade water. Peptides were eluted from columns into eight fractions using increasing concentrations of acetonitrile in aqueous triethylamine. Fractions were dried in a vacuum concentrator before reconstituting in aqueous 0.1% trifluoroacetic acid. Fractions were analyzed over 4 h acquisitions with elution from a 50 cm C_18_ EasyNano PepMap nanocapillary column using an UltiMate 3000 RSLCnano HPLC system (Thermo) interfaced with an Orbitrap Fusion hybrid mass spectrometer (Thermo). Positive ESI-MS, MS^2^ and MS^3^ spectra were acquired with multi-notch synchronous precursor selection using Xcalibur software (version 4.0, Thermo). Mascot Daemon (version 2.5.1, Matrix Science) was used to search against the *Nannochloropsis gaditana* subset of the UniProt database. To maximize the number of identified proteins, the search was conducted on a database containing concatenated data from the B31 and CCMP526 proteomes (15,363 sequences; 5,747,225 residues). The Mascot 0.dat result file was imported into Scaffold Q+ (version 4.7.5, Proteome Software) and a second search run against the same database using X!Tandem. Protein identifications were filtered to require a maximum protein and peptide false discovery rate of 3% [47] with a minimum of two unique peptide identifications per protein. Protein probabilities were assigned by the Protein Prophet algorithm [48]. Relative quantitation of protein abundance was calculated from the TMT reporter ion intensities using Scaffold Q+. TMT isotope correction factors were applied according to the manufacturer. Differentially expressed proteins were determined by applying Permutation Tests with significance levels (*p*-values) adjusted with the Benjamini–Hochberg method. TMT labelling provides sensitive measurements of differential expression of individual proteins in multiplexed samples. However, the effect of peptide length and composition means that the reporter ion responses across different proteins are only semi-quantitative estimates of abundance, i.e., different peptides/proteins have different response factors. To more accurately estimate protein quantities, the “protein abundance in multiplexed samples” (PAMUS) method [49] was applied, which is based on the empirical linear relationship between the protein abundance index (PAI) and the logarithm of absolute protein abundance [50]. The exponentially modified PAI (emPAI) for each protein was first obtained from Scaffold Q+ to estimate the relative amount of each protein in the multiplexed sample. Then for each protein, the TMT reporter ion intensities were used to quantify the proportion of emPAI attributed to each individual sample/label. The abundance of the proteins in the individual samples was then expressed in Mol% [50]. The location of mature proteins in the cell was annotated based on the “Subcellular location” field of the UniProtKB database (www.uniprot.org). Complete mass spectrometry data sets are open-access and available to download from MassIVE (MSV000085294) and ProteomeXchange (PXD018605) (doi:10.25345/C5GQ50).

### 4.6. Transcriptomics

Total RNA was extracted from cell pellets by adding 1.0 mL QIAzol (Qiagen) followed by lysis with a bead-beater (Precellys, Bertin Instruments, Montigny-le-Bretonneux, France, 0.1 mm glass beads, 6500 rpm, 15 s). After adding 0.2 mL chloroform, the sample was centrifuged (20,000 rcf, 15 min, 4 °C) and the aqueous supernatant was added directly to RNA Clean and Concentrator columns (Zymo Research, Irvine, CA, USA) and prepared according to the manufacturer instructions. The cleaned RNA was eluted from the columns using molecular grade water and quality and quantity checked using a 2200 TapeStation instrument (Agilent Genomics, Santa Clara, CA, USA) and Nanodrop Spectrophotometer (Thermo Fisher Scientific). Libraries were prepared using Poly(A) selection to enrich for mRNA and a NEBNext Ultra Directional RNA Library Prep kit for Illumina (New England Biolabs Inc., Ipswich, MA, USA) according the manufacturer protocols. Barcoded sample libraries were pooled in equal amount and sequenced on an Illumina NextSeq 500 platform using High Output Kit v2. A total of 443 million 150 bp paired-end reads were obtained and archived at NCBI web portal under Bioproject PRJNA589063.

The quality of reads was assessed with FastQC (Babraham Bioinformatics, Cambridge, UK) and gentle adapter and quality trimming (Q > 20, L > 50) was applied using cutadapt v1.13 [51]. The annotated reference genomes of *N. gaditana* were downloaded for strains CCMP526 (assembly ASM24072v1) and B31 (assembly NagaB31_1.0) and assessed. Although we used strain CCMP526 in our study (verified genetically, Appendix A), the more recent reference genome for strain B31 provided more unique mapped reads in our data (for reference comparisons see Appendix A). Our analysis therefore uses reads that were aligned to the B31 reference genome using the splice-aware aligner STAR 2.5.3a [52], with the annotation aware option. The PCR duplication rate was assessed using the Bioconductor package “dupRADAR” [53] in R v. 3.3.3 and was found to be low (<0.1%). Counts of reads for gene-level quantification were extracted using “featureCounts” [54] supplied with annotation information and strands of reads. Raw counts were imported into the Bioconductor package “DESeq2” v 1.14.1 [55] and differential expression analysis was performed with independent filtering enabled and alpha = 0.05. Genes that had an FDR *p*-adjusted value < 0.05 and L_2_fc > 1.0 (fold change of > 2) were chosen as the differentially expressed genes. Taking into account our design (four replicates in each group and fold change > 2.0) we reached more than 90% statistical power to detect differentially expressed genes [56].

### 4.7. Gene Ontology and KEGG Pathway Gene Set Enrichment Analysis

Gene ontology (GO) terms were obtained from the UniProtKB database (http://www.uniprot.org). Annotation of genes for KEGG Orthology (KO) numbers was performed using GhostKOALA [57]. Gene ontology and KEGG pathway enrichment analyses were performed for both transcriptome and proteome data sets. Gene set enrichment analysis implemented in Babelomics 5.0 suite [58], was used to detect GO functional sets of genes and proteins significantly affected by nutrient deprivation. The logistic model using the L_2_fc of all genes or proteins was employed with significance cut-off FDR-adjusted *p*-value of 0.01. GOs with log-odds ratio (LOR) <0.0 were taken to be over-represented for downregulated genes/proteins, and LOR > 0.0 were over-represented for upregulated genes/proteins. The gene set approach was also used to identify the most perturbed KEGG pathways with unidirectional changes of gene and protein expression. The analysis was performed using the Bioconductor package GAGE 2.24 [59] with L_2_fc values as per gene statistics, q < 0.05 and only pathways with more than five annotated KO numbers. GO enrichment analysis was performed separately for up- and downregulated genes using classic Fisher’s exact test in R package topGO v2.26 [60] with FDR correction at 0.05 and pruning the GO hierarchy from terms which have less than five annotated genes. To identify the most perturbated KEGG pathways with unidirectional or bidirectional changes of gene expression the gene set approach was used. The analysis was performed using the Bioconductor package GAGE 2.24 [59] with L_2_fc values as per gene statistics and only pathways with more than five annotated KO numbers.

### 4.8. Data Analysis

The protein and transcript data were associated together using their unique ID (gene, UniProt) numbers. Data was analyzed using the R programming language, and the package “nlme” [61], was used to fit a linear mixed-effects model (Figure 3b, Appendix A). The mixed-model fixed effects were (log RPKM~log Mol%) with the random effects formula (~1 + logMol%|replicate) following nlme notation, where “log RPKM” is the natural logarithm of transcript counts in units RPKM and “log Mol%” is the natural logarithm of protein abundance in Mol%. The “replicate” term is the individual turbidostat cultivation (*n* = 10). Correlation coefficients, summary statistics, and linear regression models were implemented in base R.

## 5. Conclusions

This study provides new insights into global protein and gene expression in the oleaginous microalga *Nannochloropsis gaditana*. Both proteomic and transcript sequencing methods each tended to capture the major patterns in expression, but at the whole-cell level protein and transcript associations were characteristically noisy. In *Nannochloropsis* macronutrient stress is associated with lipid remodeling and oleaginous phenotypes, but lipid metabolic processes were not highly enriched in our GO and KEGG analyses. We did however find major changes in several lipid-related proteins, including increased expression of DGAT and lipid body proteins under N-starved conditions. Pathways in lipid remodeling, fatty-acid oxidation and signaling could be prioritized for future studies, as these are key processes that determine the fate of valuable long-chain polyunsaturated fatty acids. Adjustments in respiratory/mitochondrial activity featured in our data, with shifts in TCA cycle activity and glycolytic processes providing metabolic compensation under stress. The active reshaping of organelle (compartment) proteomes and the control of inter-organelle metabolic flux are therefore important research areas. Finally, our data raises the topic of post-transcriptional mechanisms, which may in part explain the observed patterns of gene/protein/metabolite correlations.

## Figures and Tables

**Figure 1 ijms-21-06946-f001:**
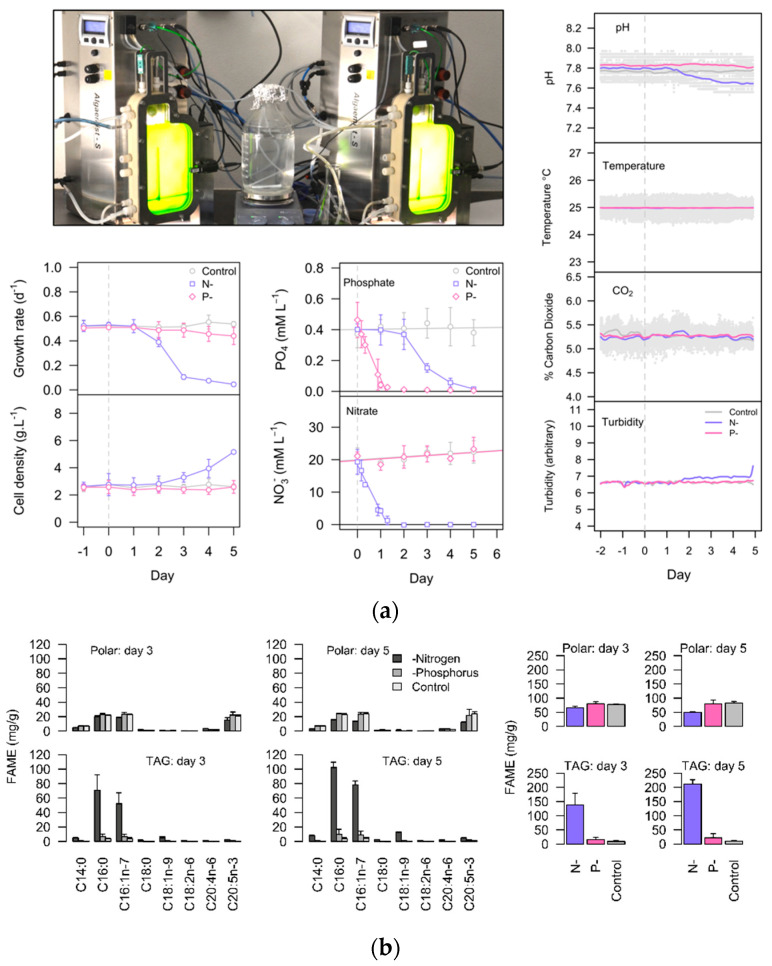
(**a**) Image of the flat-plate photobioreactors operated as turbidostats including measurement of pH, temperature, CO_2_ concentration in the sparging gas, and turbidity. The growth rate (d^−1^) and the cell density (g L^−1^) are shown with the changes in the dissolved extracellular nitrate (NO_3_^−^) and phosphate (PO_4_^3−^) concentrations (mean ± sd, *n* = 4). (**b**) Lipid analysis including the fatty-acid profiles (left) of polar and neutral lipids (TAG) in control, N-, and P- treatments after 3 and 5 days of the experiment, as fatty-acid methyl-esters—FAME (mg/g dry weight). The total FAMEs in control, N-, and P- treatments after 3 and 5 days of the experiment (right). Data are the mean ± sd of *n* = 4 experimental replicates (except *n* = 3 for N- treatments at day 5).

**Figure 2 ijms-21-06946-f002:**
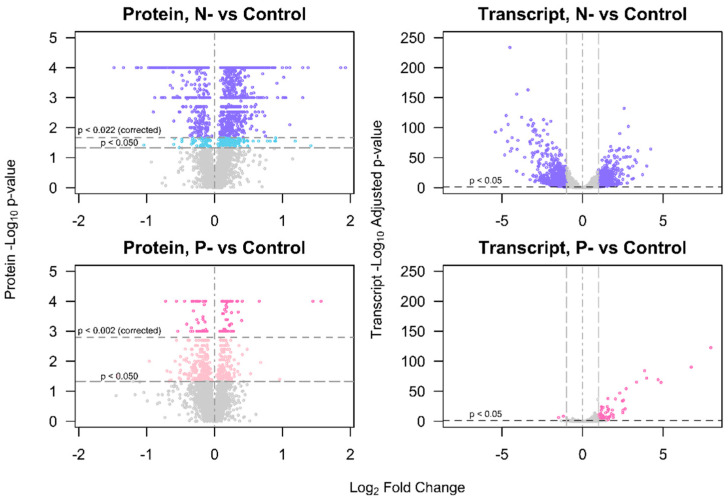
Volcano plots showing the differential expression of proteins and transcripts in the nitrogen starved (N-) and phosphorus starved (P-) treatments, vs. controls. The *x*-axis displays the log_2_ fold change (L_2_fc) of protein or transcript expression, where positive values indicate upregulated proteins and negative values correspond to downregulated proteins. The *p*-values are presented on -Log_10_ scale on the *y*-axis, and for transcripts these are the adjusted *p*-values from the DESeq2 methodology. Proteins determined significantly differently regulated at corrected thresholds *p* < 0.022 (N-/C treatments) or *p* < 0.002 (P-/C treatments) are indicated in the uppermost segment. Proteins differentially expressed at *p* < 0.050, but not reaching the adjusted threshold, are indicated in the central segment.

**Figure 3 ijms-21-06946-f003:**
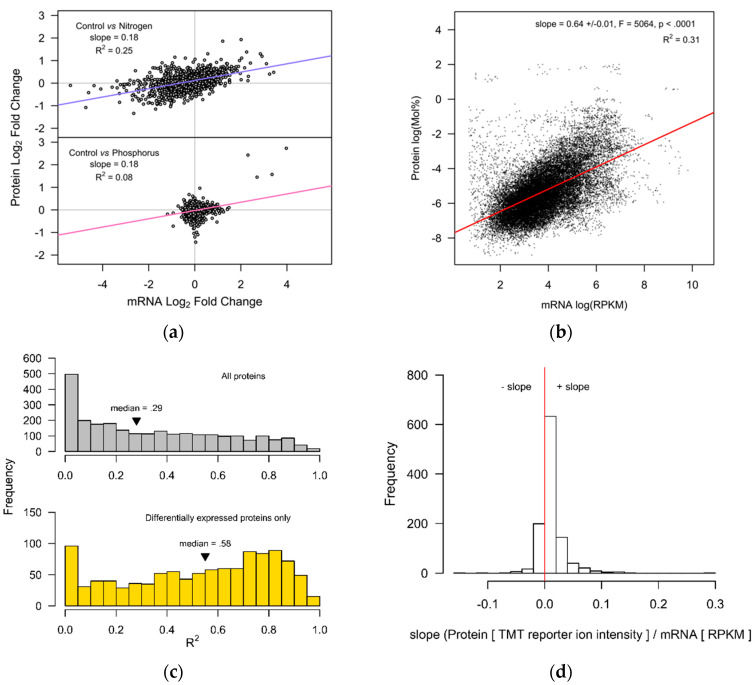
Global patterns in protein and mRNA abundance in *Nannochloropsis gaditana.* (**a**) The L_2_fc mRNA abundance vs. the L_2_fc protein abundance for N- and P- treatments, vs. controls (*n* = 2578 each). (**b**) Protein abundance in log (Mol%) vs. mRNA transcript abundance (RPKM) for all samples. The regression line was fitted with a linear mixed-effects model with random slopes and random intercepts fitted for each experimental unit (*n* = 10). Very low abundance transcripts < 2.0 RPKM were excluded. (**c**) Histograms showing the population of R^2^ values that describe the relationship between mRNA abundance (RPKM) and protein abundance (normalized TMT reporter ion intensities) for each gene/protein set. The R^2^ values are collected from *n* = 2576 linear regression models fitted separately to each gene/protein pair from the B31 genome assembly (Appendix A). The upper panel contains all of the correlations, whilst the lower panel shows only those where the proteins were significantly differently regulated (*n* = 1083), as determined by the Benjamini–Hochberg adjusted *p*-values. (**d**) The slopes showing positive or negative correlations for the same 1083 linear regression fits.

**Figure 4 ijms-21-06946-f004:**
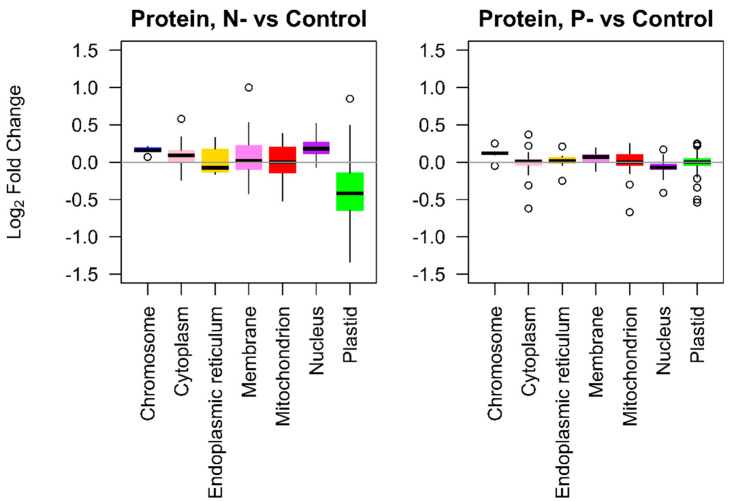
The L_2_fc of proteins localized in different subcellular compartments. Panels represent N- (*n* = 4) or P- conditions (*n* = 2), relative to the control group (*n* = 4). Annotation of locations was provided by the UniProtKB database.

**Figure 5 ijms-21-06946-f005:**
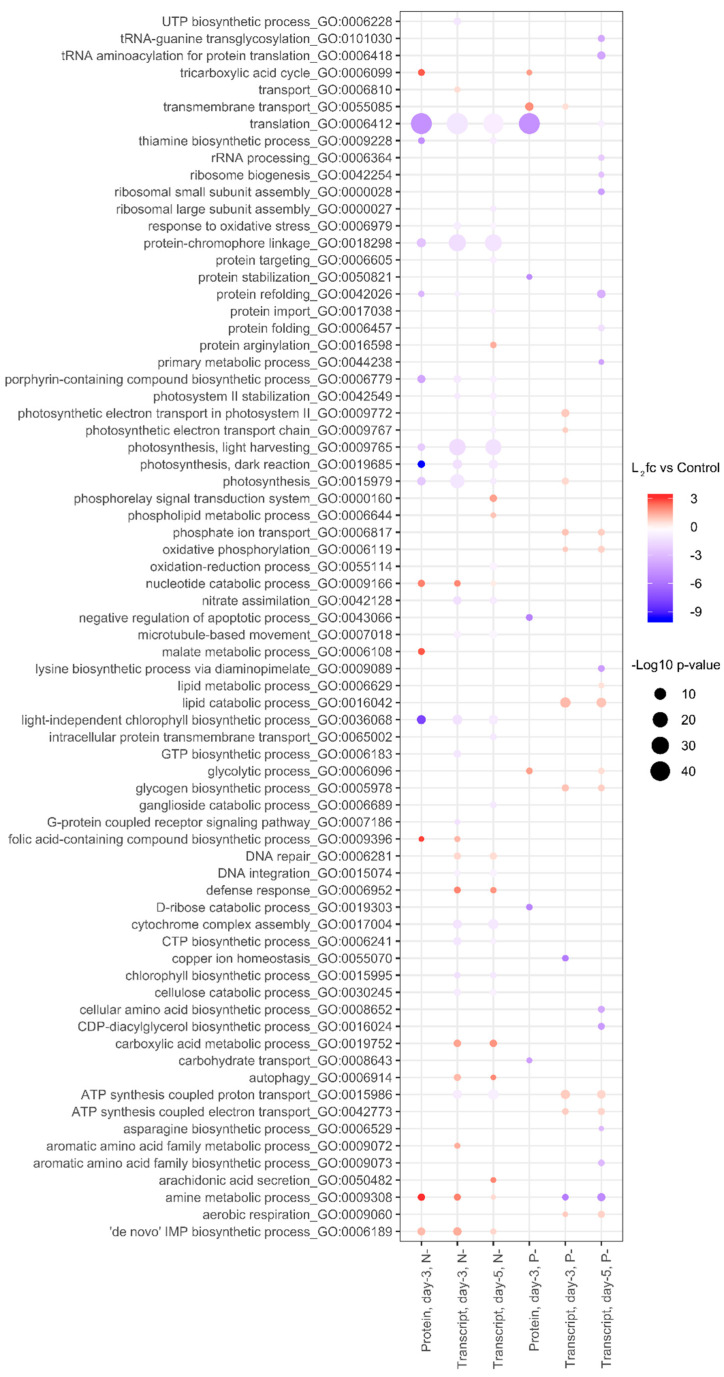
Gene set enrichment. The gene ontology classifications (GO: biological processes) of proteins and transcripts differentially expressed under nitrogen and phosphorus deprivation.

**Figure 6 ijms-21-06946-f006:**
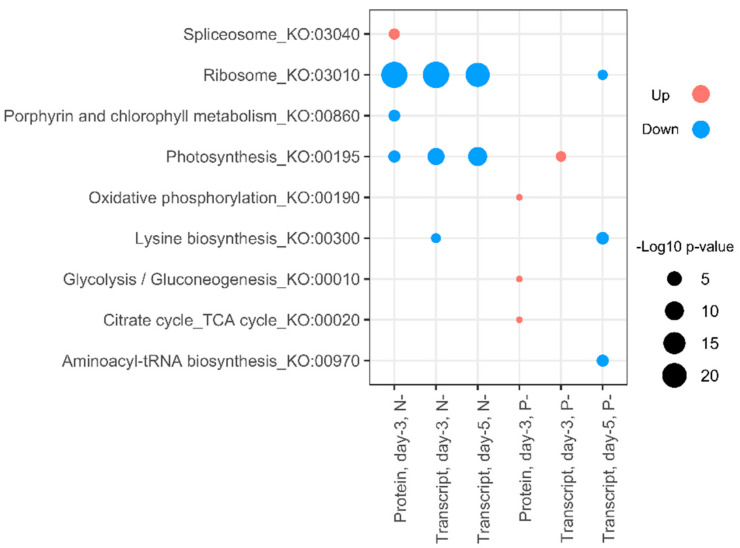
Changes in metabolic pathways. The most perturbed KEGG (KO:) metabolic pathways in the proteome and the transcriptome.

**Figure 7 ijms-21-06946-f007:**
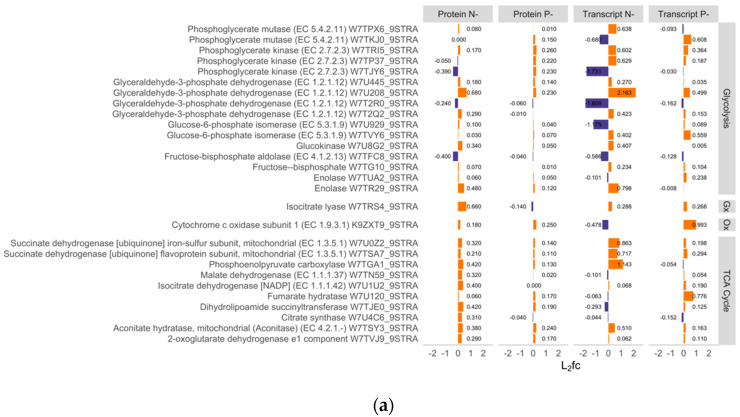
Respiratory activity under N- and P- conditions. (**a**) The L_2_fc of proteins and genes linked to glycolytic processes, the TCA cycle, glyoxylate cycle (Gx), and oxidative phosphorylation (Ox). Proteins were identified manually using GO terms and by searching for specific accessions. Transcript data were then matched to the proteins using the unique accession number. (**b**) The fold changes of the proteins and transcripts.

**Table 1 ijms-21-06946-t001:** The 30 proteins with largest fold increase and 30 proteins with the largest fold decrease in the N- treatments (*n* = 4), relative to the controls (*n* = 4). Proteins annotated as “uncharacterized” were omitted and the *p*-values are from permutation tests. The suffix string of the Accession Number “9STRA” or “NANGC” refers to the B31 or CCMP526 *N. gaditana* reference proteomes, respectively.

Rank	Identified Proteins	Accession Number	kDa	L_2_fc	*p*-Value
Upregulated					
1	Lipid droplet surface protein	W7TWF7_9STRA	18	1.93	0.0001
2	Amine oxidase	W7TFN3_9STRA	75	1.38	0.0001
3	Methylenetetrahydrofolate dehydrogenase	W7T6I6_9STRA	39	1.3	0.0001
4	Acid sphingomyelinase-like phosphodiesterase 3b	W7TQ09_9STRA	76	1.3	0.001
5	EF-Hand 1, calcium-binding site	W7TRW6_9STRA	64	1.11	0.0001
6	Lipase family protein	W7TUB0_9STRA	54	1.06	0.0001
7	Two component regulator propeller domain-containing protein	K8Z0G9_NANGC	27	1.03	0.001
8	Lipocalin protein	W7TQX7_9STRA	29	1.02	0.00021
9	Ammonium transporter	W7U477_9STRA	58	1	0.0001
10	Carbonic anhydrase, alpha-class	W7T0A1_9STRA	37	0.9	0.028
11	Cathepsin a	W7TYE0_9STRA	60	0.87	0.0001
12	Nadp-dependent glyceraldehyde-3-phosphate dehydrogenase	W7U8W3_9STRA	66	0.86	0.0001
13	Cluster of Sodium hydrogen exchanger 8	W7TNK5_9STRA	72	0.86	0.0001
14	Light harvesting complex protein	K8YPR7_NANGC	19	0.85	0.0001
15	Subfamily member 9	W7TPA4_9STRA	41	0.82	0.028
16	Plasma membrane ATPase	K8YQB4_NANGC	107	0.77	0.0001
17	Manganese lipoxygenase	W7TYD4_9STRA	73	0.77	0.0001
18	Quinoprotein amine dehydrogenase, beta chain	W7TI92_9STRA	66	0.77	0.0001
19	4-hydroxyphenylpyruvate dioxygenase	W7TNB7_9STRA	50	0.77	0.001
20	Malate cytoplasmic isoform 2	W7TPM0_9STRA	37	0.76	0.0001
21	Cluster of Violaxanthin de-epoxidase	K8YTT8_NANGC	35	0.75	0.019
22	Had-superfamily subfamily iia hydrolase	W7U270_9STRA	43	0.74	0.0001
23	Glutaryl-mitochondrial	W7TTQ4_9STRA	48	0.74	0.0001
24	Pyruvate dehydrogenase	W7TN62_9STRA	55	0.74	0.0001
25	Myotubularin-related protein 2	W7TSB4_9STRA	109	0.74	0.004
26	Cdgsh iron sulfur domain-containing protein 1	W7TPN8_9STRA	23	0.72	0.001
27	Arachidonate 5-lipoxygenase	K8Z8I5_NANGC	60	0.71	0.0001
28	Cluster of Purple acid phosphatase	W7TLQ2_9STRA	56	0.71	0.0001
29	Cluster of Expulsion defective family member (Exp-2)	K8YVZ3_NANGC	62	0.71	0.049
30	V-type proton ATPase subunit F	W7TU11_9STRA	13	0.7	0.0001
Downregulated					
30	Cytochrome p450	W7UBA8_9STRA	70	−0.77	0.0001
29	30s ribosomal protein s15	W7TEF2_9STRA	34	−0.77	0.0001
28	RNA binding s1 domain protein	W7U882_9STRA	45	−0.77	0.0001
27	Cluster of Solute carrier family 35 member b1	W7TCR9_9STRA	43	−0.77	0.7
26	Cytochrome P450 enzyme	I2CNY8_NANGC	67	−0.78	0.001
25	Heat shock protein DNAJ, cysteine-rich domain protein	W7TJ91_9STRA	13	−0.78	0.001
24	Geranylgeranyl reductase	W7THD6_9STRA	57	−0.79	0.0001
23	Coproporphyrinogen iii oxidase chloroplast	W7TZ92_9STRA	46	−0.79	0.0001
22	50S ribosomal protein L18, chloroplastic	K9ZX62_9STRA	12	−0.8	0.0001
21	50S ribosomal protein L19	K9ZV73_9STRA	14	−0.81	0.0001
20	30S ribosomal protein S9, chloroplastic	A0A023PLK7_9STRA	15	−0.82	0.0001
19	30S ribosomal protein S2, chloroplastic	K9ZWC8_9STRA	29	−0.83	0.0001
18	Nitrite reductase	W7T0E9_9STRA	46	−0.85	0.0001
17	30S ribosomal protein S8, chloroplastic	K9ZV68_9STRA	15	−0.86	0.0001
16	Cluster of H+-transporting ATPase	K8YQ29_NANGC	152	−0.87	0.0001
15	30S ribosomal protein S12, chloroplastic	K9ZVC5_9STRA	14	−0.88	0.0001
14	50S ribosomal protein L36, chloroplastic	K9ZXS5_9STRA	4	−0.88	0.001
13	Magnesium chelatase ATPase subunit I	K9ZV21_9STRA	47	−0.9	0.0001
12	50S ribosomal protein L16, chloroplastic	K9ZWF3_9STRA	16	−0.9	0.0001
11	Ribosomal protein s21	W7TSY1_9STRA	14	−0.9	0.003
10	Cluster of Mfs transporter	W7U968_9STRA	66	−0.93	0.14
9	30S ribosomal protein S17, chloroplastic	K9ZVE6_9STRA	10	−0.94	0.0001
8	30S ribosomal protein S20, chloroplastic	K9ZX69_9STRA	11	−0.94	0.0001
7	Delta 5 fatty acid desaturase	K8YSX2_NANGC	54	−0.95	0.0001
6	30S ribosomal protein S18, chloroplastic	K9ZV97_9STRA	8	−0.97	0.0001
5	Nitrate reductase	W7TAR6_9STRA	70	−1.08	0.0001
4	Ferredoxin nitrite reductase	K8YST4_NANGC	40	−1.13	0.0001
3	Light-independent protochlorophyllide reductase subunit N	K9ZV79_9STRA	50	−1.15	0.0001
2	Light-independent protochlorophyllide reductase iron-sulfur ATP-binding protein	K9ZV32_9STRA	32	−1.34	0.0001
1	NAD(P)H nitrate reductase	K8YSU6_NANGC	63	−1.48	0.0001

**Table 2 ijms-21-06946-t002:** The 30 proteins with largest fold increase and 30 proteins with the largest fold decrease in P- treatments (*n* = 2), relative to the controls (*n* = 4). Proteins annotated as “uncharacterized” were omitted and the *p*-values are from permutation tests. The suffix string of the Accession Number “9STRA” or “NANGC” refers to the B31 or CCMP526 *N. gaditana* reference proteomes, respectively.

Rank	Identified Proteins	Accession Number	kDa	L_2_fc	*p*-Value
Upregulated					
1	Sse2p	W7TMT9_9STRA	32	0.96	0.04
2	Acid sphingomyelinase-like phosphodiesterase 3b	W7TQ09_9STRA	76	0.68	0.011
3	Cluster of Calcium binding protein 39	W7T646_9STRA	51	0.61	0.59
4	Snf7 family protein	W7U1R3_9STRA	22	0.53	0.026
5	Ddi1p	W7U1J9_9STRA	41	0.52	0.97
6	Nad-dependent deacetylase	W7TT51_9STRA	38	0.48	0.004
7	Elongation of fatty acids protein	W7TSM8_9STRA	36	0.48	0.06
8	Lysyl-tRNA synthetase	W7TMK7_9STRA	20	0.45	0.13
9	Aminoglycoside phosphotransferase	W7TK75_9STRA	37	0.43	0.2
10	Pyruvate decarboxylase	K8YS66_NANGC	62	0.41	0.0001
11	Splicing arginine serine-rich 19	W7T8W4_9STRA	34	0.41	0.0001
12	Ribosomal protein	K8Z5W4_NANGC	33	0.39	0.067
13	Cluster of Trypsin family	K8Z6K0_NANGC	65	0.38	0.36
14	Cluster of Methylthioribose kinase	W7TVE0_9STRA	94	0.37	0.98
15	Ferredoxin	K8YW46_NANGC	12	0.36	0.055
16	Otu-like cysteine type protease	W7TUL0_9STRA	102	0.36	0.15
17	Protein-tyrosine low molecular weight	K8YTE7_NANGC	16	0.35	0.00023
18	Threonine aldolase	W7TQZ9_9STRA	47	0.35	0.012
19	Protein phosphatase	W7TA28_9STRA	48	0.35	0.2
20	Pre-mRNA-processing factor 17	K8Z4U6_NANGC	86	0.34	0.0001
21	Beta-ketoacyl-thiolase	W7SYP3_9STRA	8	0.34	0.022
22	Ethylmalonic encephalopathy 1	K8Z7T8_NANGC	47	0.33	0.0001
23	Soluble pyridine nucleotide transhydrogenase	W7T7X5_9STRA	17	0.32	0.14
24	Ring-finger-containing e3 ubiquitin	W7UAK3_9STRA	76	0.32	0.25
25	Glycerol kinase	W7U0M7_9STRA	24	0.3	0.072
26	Ig family protein	W7T9Y3_9STRA	60	0.3	0.082
27	Cluster of Mfs transporter	W7U968_9STRA	66	0.3	0.89
28	Mitochondrial tricarboxylate carrier family	W7TKI7_9STRA	36	0.29	0.009
29	Cdgsh iron sulfur domain-containing protein 1	W7TPN8_9STRA	23	0.29	0.025
30	NAD(P)-binding domain protein	W7TM45_9STRA	40	0.29	0.032
Downregulated					
30	Vacuolar protein-sorting-associated protein 36	W7TG31_9STRA	49	−0.44	0.34
29	Exocyst complex	W7U8I8_9STRA	115	−0.45	0.009
28	Methyltransferase type 11	W7U3Q9_9STRA	34	−0.45	0.027
27	RNA binding protein	W7TAT7_9STRA	20	−0.47	0.29
26	Light harvesting complex protein	K8YPR7_NANGC	19	−0.5	0.007
25	Diaminopimelate decarboxylase	W7TNX0_9STRA	56	−0.5	0.04
24	DNA polymerase subunit Cdc27	W7TMW3_9STRA	62	−0.51	0.013
23	Tubulin-tyrosine ligase-like protein	W7TWY1_9STRA	79	−0.51	0.024
22	Translocase of inner mitochondrial membrane 50-like protein	K8YTV0_NANGC	43	−0.51	0.13
21	Cluster of Protease do-like 9	W7TU24_9STRA	69	−0.52	0.61
20	TatA-like sec-independent protein translocator subunit	W7T3A7_9STRA	22	−0.54	0.001
19	Photosystem II reaction center protein H	K9ZXQ7_9STRA	7	−0.54	0.084
18	Cyclic nucleotide-binding protein	W7TMP7_9STRA	25	−0.56	0.002
17	Ubiquilin	I2CQX3_NANGC	47	−0.6	0.07
16	Ribokinase	W7TXK5_9STRA	34	−0.62	0.21
15	Ankyrin	W7TWU3_9STRA	48	−0.63	0.068
14	Soluble nsf attachment protein receptor	W7TW41_9STRA	32	−0.66	0.27
13	Elongation of fatty acids protein	W7U1Y8_9STRA	37	−0.67	0.098
12	Anamorsin homolog	W7TKP2_9STRA	30	−0.67	0.19
11	Adenylate kinase	K8ZCS9_NANGC	19	−0.68	0.1
10	Mitochondrial carrier domain protein	W7TRC0_9STRA	50	−0.68	0.13
9	Set domain protein	W7TKH2_9STRA	119	−0.73	0.75
8	ATP-dependent RNA helicase DDX23/PRP28	K8YWH1_NANGC	91	−0.77	0.13
7	Pentatricopeptide repeat-containing protein	W7TSL2_9STRA	138	−0.81	0.24
6	Fgd6 protein	K8Z5M8_NANGC	33	−0.84	0.26
5	Polypyrimidine tract binding protein	I2CQY0_NANGC	35	−0.96	0.01
4	Major facilitator superfamily	W7UAL7_9STRA	66	−0.99	0.21
3	U3 small nucleolar RNA-associated	W7UBP4_9STRA	207	−1.1	0.049
2	Phytanoyl-dioxygenase	W7T3Z1_9STRA	24	−1.2	0.047
1	DNA damage-binding protein 1a	I2CQY4_NANGC	41	−1.45	0.14

**Table 3 ijms-21-06946-t003:** Summary of the experimental samples used for proteomic and transcriptomic analysis. The “No. Cultivations” is the total number of replicate turbidostat cultures available for each treatment. Ten proteome samples were obtained after 3 days of C, N-, or P- treatment. Twelve RNA samples were obtained after both 3 and 5 days and are repeated measurements from the same experimental units.

		Day 3		Day 5	
Treatment	No. Cultivations	Protein	Transcript	Protein	Transcript
Control (C)	4	4	4	−	4
Nitrogen (N-)	4	4	4	−	4
Phosphorus (P-)	4	2	4	−	4
Total	12	10	12	−	12

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
