# Peer review of "Proteomic and Transcriptomic Patterns during Lipid Remodeling in Nannochloropsis gaditana"

_ijms, 2020, doi:10.3390/ijms21186946_

Round 1

Reviewer 1 Report

Review  result of ijms-908418 by Hulatt et al.

The different responses of marine microalga Nannochloropsis gaditana to N and P deficiency have investigated using proteomic and transcriptomic analysis. The results reveal the responses in gene expression (mRNA and protein) are different in several metabolisms, particularly TAG synthesis. 

The methods are solid and the sample has 4 replicates. However, several points need to be clarified.

  1. Why the data from day 3 and day 5 are chosen for the analysis? According to Figure 1, the medium P and N contents have been already decreased since 3 days. There is no difference between day 3 and day 5. Therefore, authors need to explain why use day 3 and day 5 samples for the comparison in protein and gene expression. Why not use day 1 and day 5, for example? 
  2. Figure 1 shows that the cell growth rate is not significantly depressed by -P treatment, although the cellular P content has been decreased after 3 days of culture. Please explain it and compared to other studies.
  3. Why te proteomic data were not analyzed on day 5 for the -P treatment? 
  4. In the results, the difference between day 3 and day 5 is not clearly described. It s hard to follow the responses between early period (day 3) and long-tern treatment (day 5).
  5. The statistical analysis should be done for the results of Figure 1. Please indicate in the Materials and Methods.
  6. Transcription factor is important in the regulation of gene expression. The TF should be included in the analysis.   

Accordingly, this article needs a major revision before it can be accepted for publication.

Author Response

Thank you very much for reviewing our article, please find below our responses to the questions raised.

The methods are solid and the sample has 4 replicates. However, several points need to be clarified.

1. Why the data from day 3 and day 5 are chosen for the analysis? According to Figure 1, the medium P and N contents have been already decreased since 3 days. There is no difference between day 3 and day 5. Therefore, authors need to explain why use day 3 and day 5 samples for the comparison in protein and gene expression. Why not use day 1 and day 5, for example?

We chose day 3 for the combined protein and gene measurements as the mid-point in the application of nutrient stress. Day 5 was the practical end-point of the experiment.

The maximum length of the experiment was constrained by the cessation of growth in the N- treatments (5 days), where it became hard to maintain the cultures at constant density after this point. By 5 days, the N- cells also became difficult to extract and clean the protein/transcript samples presumably due to abundant lipids and carbohydrates and low protein contents. Taking 5 days as the practical endpoint for the experiment, we chose day 3 as the mid-point for the main protein/transcript measurements because it allowed sufficient time for the onset of nutrient deprivation to take place (Figure 1). In contrast, sampling at day 1 would have reduced the effect-size of nutrient stress and reduced our ability to detect changes in the cells.

We have added some information to the methods section where the sampling is described (L488-): “ The maximum duration of our experiment was 5 days, by which time growth in N- treatments had nearly ceased and the limit of turbidity control was reached. Based on the cultivation data in Figure 1 we selected day 3 for proteomics and transcriptomics analysis, because it represented the mid-point in the onset of stress conditions, allowing sufficient time to detect metabolic and molecular changes in the cells “

2. Figure 1 shows that the cell growth rate is not significantly depressed by -P treatment, although the cellular P content has been decreased after 3 days of culture. Please explain it and compared to other studies.

It’s a good point that the P- conditions did not cause immediate reduction in growth rate compared with the N- treatments and we did not describe this fully. We can explain this in terms of two features of cell phosphorus requirements that fundamentally differ from the cell nitrogen requirement. First, in phosphorus replete conditions (pre-treatment), luxury phosphorus uptake takes place and many algae species may accumulate excess intracellular P that can initially compensate during subsequent phosphorus-deprived conditions. Second, unlike the nitrogen required for large-scale protein synthesis, phosphorus is found in molecules (e.g. certain lipids) that can be functionally substituted and replaced under -P conditions, meaning that the effects of phosphorus deficiency can be further dampened.

To address this, we have added the following to the manuscript:

In the results we have added two sentences describing the different effects of N- and P- on the growth rates (L93-): “ The N- cultures experienced an immediate reduction in growth to 0.11 +-0.02 d-1 at day 3 and 0.05 +-0.02 d-1 at day 5. In comparison the onset of P- conditions was more dampened with the growth rate 0.49 +-0.06 d-1 at day 3 and 0.44 +- 0.07 d-1 at day 5.”

In the discussion section 3.2 we have added (L381-) “In our data the dampened onset of P- effects contrasted with the rapid reduction in growth and changes in protein/gene expression observed in N- conditions. These differences can be rationalized by the way phosphorus is metabolized inside the cell. Under nutrient-replete conditions luxury phosphorus uptake takes place and cells can accumulate excess reserves of intracellular P, which acts as a short-term buffer during P- conditions [32]. Secondarily, certain classes of phosphorus-containing compounds can be functionally replaced by phosphorus-free compounds (see section 3.3), which reduces the impact of P- conditions on metabolism.”

Including the following reference:

Sforza, E.; Calvaruso, C.; La Rocca, N.; Bertucco, A. Luxury uptake of phosphorus in Nannochloropsis salina: effect of P concentration and light on P uptake in batch and continuous cultures. Biochemical Engineering Journal 2018, 134, 69-79.

3. Why te proteomic data were not analyzed on day 5 for the -P treatment?

It would be interesting to measure changes in the proteome at different time points, but the proteomic data were unfortunately limited by cost and the maximum number of replicates (ten) that are possible to multiplex on an LC-MS/MS run with the TMT labelling method.

4. In the results, the difference between day 3 and day 5 is not clearly described. It s hard to follow the responses between early period (day 3) and long-tern treatment (day 5).

Due to the different factors (treatments, time points) it is a bit challenging to describe all the features but keep the manuscript concise. We wanted to focus on the proteomics (day 3) and the comparison between the proteomic and transcriptomic data as the most novel aspects. We think that by adding further description the manuscript would become longer at the expense of readability and detract from the more interesting parts of the proteomics data.

5. The statistical analysis should be done for the results of Figure 1. Please indicate in the Materials and Methods.

Figure 1 contains the cultivation data and fatty acid profiles. We could indeed conduct statistical tests of these variables, but we think it would make the text description more difficult to read and we would prefer to keep this initial part of the results as straightforward as possible. We do not think that adding statistics on the cultivations would enhance the interpretation of the proteomics/transcripts results.

6. Transcription factor is important in the regulation of gene expression. The TF should be included in the analysis.

Unfortunately we do not have the transcription factor (TF) data.

Reviewer 2 Report

ijms-908418

Title: Proteomic and transcriptomic patterns during lipid remodeling in Nannochloropsis gaditana

The current research article evaluates the effects of bioavailable nitrogen and phosphorus deprivation on the proteome and transcriptome of the oleaginous marine microalga Nannochloropsis gaditana.

Overall, the manuscript is well written and detailed information was provided. However, the conclusion section is very short and not supported with data, please revise conclusion section.  Also, the language and grammatical correction required.

Author Response

Thank you very much for reviewing our article

1. Overall, the manuscript is well written and detailed information was provided. However, the conclusion section is very short and not supported with data, please revise conclusion section.  Also, the language and grammatical correction required.

We would prefer to keep the conclusions section brief to avoid repeating elements of the results and discussion, and would like to focus on drawing the different threads in the manuscript together.

Reviewer 3 Report

This manuscript provides an integrated and comparative overview of Nannochloropsis gagitana proteomes and transcriptomes under phosphorous and nitrogen deprivation conditions.
The manuscript is very well-written; methods and results are described in great details.
Although comparative studies of N and P deprivation responses were performed with some microalgae, this study has utilized quantitative proteomics and transcriptomics and several statistical methods to establish global patterns in protein and mRNA abundance. This study further supports the distinct remodeling of cellular metabolism under deprivation of two key nutrients. 

I have only minor comments to this manuscript.
line 94 Please amend lipid phenotypes to lipid species. Overall, only fatty acid profiling was performed, then is better to use common terminology.
Please explain how subcellular proteomes were established .
Please, cross-check the manuscript, including Tables, for typos, in many instances letters are not capitalized.

Author Response

Thank you very much for reviewing our article, please find below our responses to the questions raised.

This manuscript provides an integrated and comparative overview of Nannochloropsis gagitana proteomes and transcriptomes under phosphorous and nitrogen deprivation conditions.
The manuscript is very well-written; methods and results are described in great details.
Although comparative studies of N and P deprivation responses were performed with some microalgae, this study has utilized quantitative proteomics and transcriptomics and several statistical methods to establish global patterns in protein and mRNA abundance. This study further supports the distinct remodeling of cellular metabolism under deprivation of two key nutrients. 

I have only minor comments to this manuscript.

  1. line 94 Please amend lipid phenotypes to lipid species. Overall, only fatty acid profiling was performed, then is better to use common terminology.

Corrected, replaced with “fatty-acids” and other instances in the manuscript.

  1. Please explain how subcellular proteomes were established.

The subcellular proteomes were established based on the “Subcellular location” field of UniProtKB protein accessions for Nannochloropsis gaditana (from the same CCMP526 and B31 genomes used in the proteomics database search). UniProtKB encompasses both manually-curated (Swiss-Prot) and automatically annotated (TrEMBL) datasets and the data are traceable.

To address this we have added an extra statement to the methods section (L582-) “ The location of mature proteins in the cell was annotated based on the “Subcellular location” field of the UniProtKB database (www.uniprot.org). ”

  1. Please, cross-check the manuscript, including Tables, for typos, in many instances letters are not capitalized.

We have checked the manuscript and made a few changes in Tables 1 and 2.

Round 2

Reviewer 1 Report

Review result of ijms-908418 by Hulatt et al.

According to the responses, the comments are considered, whereas some are not accepted by authors. I can accept their defense for why some points, such as statistics was not indicated in Figure 1. I still suggest a short description on statistics is needed.

Furthermore, because the transcriptome and proteome data should be deposited in the public databank, their data should be submitted to the public databank, such as NCBI and give the ID number in the manuscript. 

Author Response

Many thanks for reviewing our manuscript. We have made some minor changes as requested.

> According to the responses, the comments are considered, whereas some are not accepted by authors. I can accept their defense for why some points, such as statistics was not indicated in Figure 1. I still suggest a short description on statistics is needed.

-- we have added some description of the statistics to section 4.8 and Figure 1 legend

> Furthermore, because the transcriptome and proteome data should be deposited in the public databank, their data should be submitted to the public databank, such as NCBI and give the ID number in the manuscript.

-- we have fully released the proteomics data. the doi and links to the proteomics and transcript data can be found in the methods section and in the Supplementary Materials section at the end of the manuscript.